# A Comparative Assessment of High-Throughput Quantitative Polymerase Chain Reaction versus Shotgun Metagenomic Sequencing in Sediment Resistome Profiling

Nazima Habibi [1,*] , Saif Uddin [1,*] , Montaha Behbehani [1], Hanan A. Al-Sarawi [2], Mohamed Kishk [1] , Waleed Al-Zakri [1], Nasreem AbdulRazzack [1], Anisha Shajan [1] and Farhana Zakir [1]

[1] Environment and Life Science Research Centre, Kuwait Institute for Scientific Research, Jamal Abdul Nasser Street, Safat 13109, Kuwait; wzakri@kisr.edu.kw (W.A.-Z.); ashajan@kisr.edu.kw (A.S.)

[2] Environment Public Authority, Fourth Ring Road, Shuwaikh Industrial 70050, Kuwait; h.alsarawi@epa.gov.kw

* Correspondence: nhabibi@kisr.edu.kw (N.H.); sdin@kisr.edu.kw (S.U.)

**Abstract:** Prolonged and excessive use of antibiotics has resulted in the development of antimicrobial resistance (AMR), which is considered an emerging global challenge that warrants a deeper understanding of the antibiotic-resistant gene elements (ARGEs/resistomes) involved in its rapid dissemination. Currently, advanced molecular methods such as high-throughput quantitative polymerase chain reaction (HT-qPCR) and shotgun metagenomic sequencing (SMS) are commonly applied for the surveillance and monitoring of AMR in the environment. Although both methods are considered complementary to each other, there are some appreciable differences that we wish to highlight in this communication. We compared both these approaches to map the ARGEs in the coastal sediments of Kuwait. The study area represents an excellent model as it receives recurrent emergency waste and other anthropogenic contaminants. The HT-qPCR identified about 100 ARGs, 5 integrons, and 18 MGEs (total—122). These ARGs coded for resistance against the drug classes of beta-lactams > aminoglycoside > tetracycline, macrolide lincosamide streptogramin B (MLSB) > phenicol > trimethoprim, quinolone, and sulfonamide. The SMS picked a greater number of ARGs (402), plasmid sequences (1567), and integrons (168). Based on the evidence, we feel the SMS is a better method to undertake ARG assessment to fulfil the WHO mandate of "One Health Approach." This manuscript is a useful resource for environmental scientists involved in AMR monitoring.

**Keywords:** antimicrobial resistance; antibiotic resistance genes; plasmids; mobile genetic elements; next-generation sequencing; quantitative polymerase chain reaction

## 1. Introduction

Antibiotics are pharmaceuticals that find daily usage in the healthcare and agriculture sectors to fight against bacterial infections. The prolonged use of certain drugs has resulted in the evolution of antibiotic resistance within the organisms and is expressed as antibiotic resistance genes (ARGs) [1]. The thresholds are alarming, with approximately 4.95 million fatalities in 2019 [2] that is expected to reach 10 million by 2050 [3]. A pressing concern among the World Health Organization (WHO) and the Centre for Disease Control and Prevention (CDC) is to design applicable surveillance strategies [4] to understand its spread and mitigation across the environment. The WHO, jointly with the International Monetary Fund (IMF) and the World Bank (WB), has proposed the "One Health and Global Health" approach with the primary objective of monitoring the environmental health of humans and animals [5,6]. Resistome profiling is thus a holistic advancement, surveying the origin and richness of antibiotic resistance genes (ARGs) as well as the underlying factors, i.e., mobile genetic elements (MGEs) promoting its dissemination and evolution [6].

Antibiotics are commonly used to treat human and animal infections. Among these beta-lactams are the last choice drugs and are used to treat urinary, respiratory, and blood infections [7]. The bactericidal properties of its several subcategories including penicillin, cephalosporins, carbapenems, amoxicillin, cefazolin, and meropenem are now being compromised due to the development of ARGs [8]. Some other bactericidal antibiotics aminoglycosides, fluoroquinolones, and glycopeptides need to be investigated for respective ARGs. Numerous antibiotics possess bacteriostatic properties such as glycylcycline, tetracyclines, lincosamide, macrolides, oxazolidines, and sulfonamides [8]. There is a dire need to check for the ARGs against these antibiotic classes and the risk they pose to human, animal, and environmental subjects.

Historically, antibiotic resistance monitoring has been practiced for the examination of targeted pathogens and the genes they carry mostly in clinics [9–13], agriculture [14–16], and water-treatment facilities [17–19]. The ARGs bearing ARBs from these situations are channelized into the nearby geographic locations that often become the point source of transmissible outbreaks [20–22]. Environmental reconnaissance has thereafter gained importance in the last few decades [23–25]. Over the years, it has been realized that the diversity and complexity of an environmental habitat is unmatched and therefore following the traditional practices might limit our understanding of the ecological resistome. A universal approach is desirable to concurrently identify both the infectious and non-infectious carriers present within a system. Several molecular assays were developed and a high-throughput quantitative polymerase chain reaction (HT-qPCR) has been emphasized as one of the most efficient genome-centric approaches to map the entire resistome of a territory [26–28].

The popularity of HT-qPCR is due to its ability to capture environmental concentrations of ARGs in real time. The multiplex primers add more benefit by mapping hundreds of genes, mobile genetic elements (plasmids and class 1, 2, and 3 integrons), 16S rRNA genes (bacterial copies), and the WHO-recommended taxonomic groups simultaneously in a single run [26,29]. Although widely used, the HT-qPCR is limited in picking low abundance genes (>10 copies) and unknown genes. The environmental samples are very challenging and getting reasonable quantifiable material from these settings is difficult. This is recently being compensated for with the use of next-generation sequencing.

The shotgun metagenomic (SMS) method is nowadays considered a gold standard in terms of mapping the entire resistome [30–40]. Not only does it provide information on all the genes present within the milieu but it also captures cryptic genes. Once the sequence data are available, taxonomic, functional, and resistome profiling can be conducted simultaneously. Higher sequencing coverage allows the filtering of low-abundance genes. Researchers have started applying SMS [37,38,41], but it is relatively expensive and requires high bioinformatics skills to derive meaningful conclusions. However, owing to its popularity, regular technical upgrades are being conducted, and several new platforms are being made available. The cost is also getting reduced with time.

To compare the efficacy of the two methods, in the present investigation, we employed the high throughput quantitative PCR (HT-qPCR) method to identify selected antibiotic-resistant genes and the associated mobile genetic elements from marine sediments receiving effluent discharges along the urban coast of Kuwait. We also compared the output of the HT-qPCR with the results of the shotgun metagenomic sequencing (SMS).

## 2. Materials and Methods

### 2.1. Study Area and Sample Collection

The coast of Kuwait is well developed, with numerous emergency outfalls and stormwater outlets connecting to the sea. The presence of pharmaceuticals in seawater, especially antibiotics, poses a significant threat to marine biota near these outfalls [42]. Antibiotics such as sulfamethoxazole, ciprofloxacin, clarithromycin, cefalexin, erythromycin, azithromycin, tetracycline, ofloxacin, trimethoprim, dimetridazole, metronidazole, metronidazole-OH, and ronidazole were found in the influent and effluent wastewaters of

Kabd and Um Al Hayman [42]. Our recent investigation showed that the pharmaceuticals are not restricted to the wastewater stream but are present in seawater samples collected across the Kuwait coastal area [43]. We, therefore, used the marine surface sediments of Kuwait as a model system in the present investigation. Briefly, grab samples (covering 10–15 cm sediment profile) were added to 50 mL sterile tubes (Corning®, Corning, AZ, USA). Samples were shifted to Kuwait Institute for Scientific Research (KISR) laboratories in ice and aliquoted to store at −20 °C until DNA extraction. Detailed sample collection and site specifications are described elsewhere [44].

### 2.2. DNA Isolation

DNA was isolated from the sediment samples employing the Qiagen power soil kit (Qiagen, Germantown, MD, USA). A total of a 0.250 mg sample was weighed and placed in a 1.5 mL sterile Eppendorf tube. The sediment was resuspended in 60 μL of solution C1 and was transferred to the power bead tube secured to a horizontal vortex adapter and vortexed at maximum speed for 10 min. The lysate was pelleted via centrifugation at $10,000 \times g$ for 30 s. The supernatant was transferred to a clean 2 mL collection tube, followed by the addition of solution C2, incubation at 2–8 °C for 5 min, and centrifugation at $10,000 \times g$ for 1 min. The supernatant was treated with solution C3 (200 μL) to remove inhibitory substances. The dissolved DNA was bound to the MB spin columns by the addition of solution C4 (1200 μL). The C5 solution was used twice to wash the DNA and final elution was performed in solution C6 (10 μL). Multiple aliquots ($n = 5$) from the same site were used for DNA extraction and pooled to reach the desired concentrations. The quantity of isolated was estimated using a Qubit fluorometer (Thermo Fisher Scientific, Carlsbad, CA, USA) employing the HS dsDNA Qubit assay kit (Invitrogen, Carlsbad, CA, USA). The intactness of bands was viewed on a 0.8% agarose gel run at 100 V/cm for 45 min (Bio-Rad, Hamburg, Germany).

### 2.3. SmartChip[TM] HT-qPCR Analysis

DNA isolated from the above step was freeze-dried and transported to Resistomap Oy (Helsinki, Finland) for the HT-qPCR analysis. The DNA samples were resuspended in nuclease-free water, and 2 ng/μL was added to the PCR reaction mix containing $1 \times$ SmartChip [TM] green gene expression master mix, 300 nM of primers, and nuclease-free water. The primers ($n = 296$) comprised 268 oligos for ARGs, 8 for integrons, and 20 for other genes [27]. The ARG oligos were chosen to cover the ARGs against the drug classes of beta-lactams, aminoglycosides, tetracyclines, fluoroquinolones, MLSB, tetracycline, vancomycin, phenicol, trimethoprim, and sulfonamides. A total reaction volume of 100 nL ($n = 3$ replicates) was loaded on a SmartChip[TM] ($n = 5184$ wells) employing an automated SmartChip[TM] multisample nanodispenser (TakaraBio, San Jose, CA, USA). PCR was conducted on a SmartChip[TM] Real-Time PCR System (TakaraBio, San Jose, CA, USA) through a PCR program including initial enzyme activation at 95 °C for 10 min followed by 40 cycles of denaturation at 95 °C for 30 s and annealing at 60 °C for 30 s. A melt curve analysis was performed using the SmartChip[TM] qPCR software to exclude false positive data (amplicons with unspecific melting curves or multiple peaks). The cycle threshold ($C_T$) was set at 27. The average $C_T$ (triplicates) was used for estimating the relative abundance with reference to the 16S rRNA gene employing the universal $2^{-\Delta CT}$ method ($\Delta C_T = C_T$ detected gene—$C_T$ 16S rRNA gene) [45]. The 16S rRNA gene also served as a positive control. A negative control was run separately with sterile nuclease-free water to check for cross-contamination.

### 2.4. Shotgun Metagenomic Sequencing

The same set of samples were subjected to whole genome metagenomic sequencing at Novogene Ltd. Taipei City, Taiwan. The steps of analysis for this method have been described by Habibi et al. [44]. Shortly, 1 μg of DNA was sonicated, end-repaired, A-tailed, indexed, and sequenced on an Illumina NovaSeq 6000 platform following the

2 × 150 bp paired-end chemistry. The raw sequences were processed through standard bioinformatics pipelines for quality check, N-removal, and adapter trimming. High-quality sequences were assembled into ≥500 bp scaftigs through MEGAHIT v 1.04. The scaftigs were aligned in MetaGeneMark v 2.10 for gene prediction. Genes were referenced in the Comprehensive Antibiotic Research Database (CARD) (BLASTP e value ≤ 1 × 10) to filter antibiotic resistance genes (ARGs). Plasmids, integrons, and insertion sequences were picked through alignment with integral, isfinder, and plasmid databases v 2018 ($-e\,1 \times 10^{-10}$, BLASTN), respectively [46].

*2.5. Statistical Analysis*

Data visualization and statistical analysis were performed by uploading the gene profiles on the ResistoXplorer web tool [47]. Microsoft Excel v2013, Numbers 12.1 (Macintosh HD, Apple Inc., Cupertino, CA, USA) and box plot maker (Statistics Kingdom) [48] were used for drawing bar charts, donut plots, and box plots, respectively.

**3. Results**

*3.1. HT-qPCR Analysis*

3.1.1. Bacterial Gene Copies

The 16S rRNA primers in the HT-qPCR panel yielded positive profiles for each sample, indicating the presence of bacterial DNA at each sampling site. The 16S rRNA gene copies exhibited spatial variations (Figure 1). The maximum number of gene counts were present at S6 (mean = 7log), followed by S12 and S1 (mean = 5log). The gene copies were below 4log at S2, S3, S4, S5, S7, and S8. The bacterial cell counts at S9, S10, and S12 were very low.

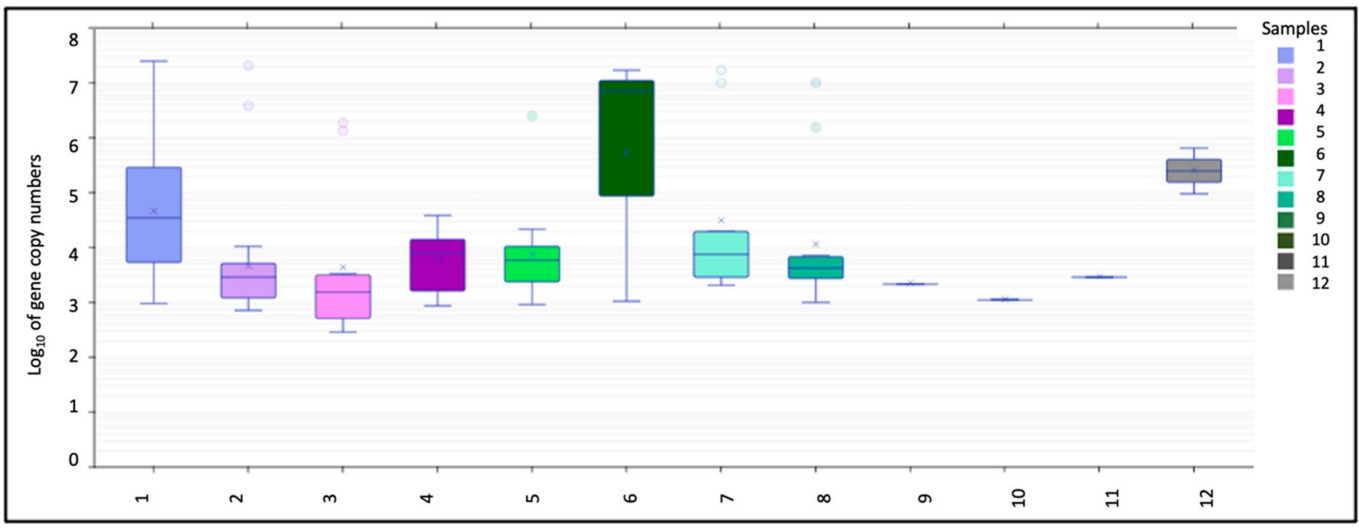

**Figure 1.** 16S rRNA gene copies found in the marine sediments of Kuwait as revealed using the HT-qPCR assay.

3.1.2. Antibiotic Resistance Genes and Drug Classes

A total of 122 antibiotic-resistant gene elements (100 ARGs, 5 integrons, and 18 MGEs) were present in the 12 sediment samples collected from Kuwait's shoreline. The genes originated from seven major drug classes such as aminoglycoside, beta-lactam, phenicol, sulfonamide, tetracycline, trimethoprim, macrolide lincosamide streptogramin B (MLSB), quinolone, and others. The most abundant were beta-lactams (23%) with 25 gene IDs namely $bla_{TEM\_1}$, $bla1$, $bla_{CARB}$, $bla_{GES}$, $cfxA$, $bla_{OXA58\_2}$, $bla_{ROB}$, $bla_{VEB}$, $cphA\_1$, $bla_{MIR}$, $cphA\_2$, $bla_{CMY\_2}$, $bla_{SHV11}$, $bla_{OXA10\_1}$, $bla_{CMY\_1}$, $bla_{MOX}/bla_{CMY}$, $bla_{BEL-nonmobile}$, $ampC\_4$, $bla_{CMY2}$, $ampC\_1$, $ampC\_6$, $bla_{SHV\_2}$, $bla_{ACT\_2}$, $bla_{PER\_1}$, and $bla_{ACT}$. This drug class was followed by aminoglycoside (21%), with 16 gene IDs such as $aac6\text{-}aph2$, $aadA6$, $aadB$, $aadA2\_3$, $aadA\_1$, $strB$, $aadA16$, $aadA1\_2$, $aadA2\_1$, $aac(6')\text{-}lb\_1$, $aadA5\_2$, $aadA10$, $aac(6')\text{-}II$, $aacC2$, $aadA7$,

and *aphA3_1*. Tetracycline (*tetPA*, *tetA_2*, *tetX*, *tetW*, *tetM*, *tetQ*, *tetG*, *tetO_2*, *tet32*, *tet39*, *tet44*, *tetL_2,* and *tetE*) and MLSB (*InuF*, *mefA_1*, *mefA*, *ermB_2*, *ereA*, *InuC*, *ermB_3*, *mefB*, *ermF*, *erm42*, *mphA*, *IsaC,* and *msrE)* were next in order with 13 genes (12%) each. A total of eight genes (7%) were detected in the drug class of phenicol (*cmlA_2*, *catQ*, *cmlA_4*, *catB3*, *catB8*, *catB2*, *mdtL,* and *floR_1*). Six genes (5%) were confirmed in trimethoprim (*dfrA1_1*, *dfrA15*, *dfrB*, *dfra17*, *dfrA27*, and *dfrA12*) and quinolone (*qnrS2*, *qnrVC1_VC3_VC6*, *qnrS_1*, *qnrB_2*, *qnrB*, and *qnrD*) each. In sulfonamide 5 genes (3%), *sul1_1*, *sul1_2*, *sul2_1*, *sul2_2*, and *sul4* were recorded. About eight genes (7%) *merA*, *qacEΔ1_3 qacEΔ1_1*, *arr2*, *crAss64*, *crAss56*, *sat4,* and *mcr1* were classified as others. Mobile genetic elements (*tnpA_1*, *tnpA_6*, *tnpA_2*, *ISPps*, *ISI247_2*, *tnpA_3*, *IncP_oriT*, *tnpA_5*, *ISI247_1*, *tnpA_4*, *IS613*, *IncN_rep*, *Tp614*, *ISAba3*, *IncQ_oriT*, *orf37-IS26*, and *Tn5*) and integrons (*intl_3*, *intI1_2*, *intI1_1*, *intl3*, and *intI3_2*) were also found along with the ARGs resistant to above drug classes. All the ARGs are listed in Table 1. The relative abundance of each gene ID is given in Table S1.

**Table 1.** Antibiotic-resistant gene elements detected using HT-qPCR in marine sediments collected from Kuwait.

| Sulphonamides | Tetracycline | Beta-Lactam | Other | Aminoglycoside | Quinolone | MLSB | Trimethoprim | Phenicol | Integrons | MGEs |
|---|---|---|---|---|---|---|---|---|---|---|
| sul1_1 | tetPA | $bla_{TEM\_1}$ | merA | aac6-aph2 | qnrS2 | lnuF | dfrA1_1 | cmlA_2 | Intl_3 | tnpA_1 |
| sul1_2 | tetA_2 | bla1 | qacEΔ1_3 | aadA6 | qnrVC1_VC3_VC6 | mefA_1 | dfrA15 | catQ | intlI_2 | tnpA_6 |
| sul2_1 | tetX | $bla_{CARB}$ | qacEΔ1_1 | aadB | qnrS_1 | mefA | dfrB | cmlA_4 | intlI_1 | tnpA_2 |
| sul2_2 | tetW | $bla_{GES}$ | arr2 | aadA2_3 | qnrB_2 | ermB_2 | dfra17 | catB3 | intl3 | ISPps |
| sul4 | tetM | cfxA | crAss64 | aadA_1 | qnrB | ereA | dfrA27 | catB8 | Intl3_2 | ISI247_2 |
| | tetQ | $bla_{OXA58\_2}$ | crAss56 | strB | qnrD | lnuC | dfrA12 | catB2 | | tnpA_3 |
| | tetG | $bla_{ROB}$ | sat4 | aadA16 | | ermB_3 | | mdtL | | IncP_oriT |
| | tetO_2 | $bla_{VEB}$ | mcr1 | aadA1_2 | | mefB | | floR_1 | | tnpA_5 |
| | tet32 | cphA_1 | | aadA2_1 | | ermF | | | | ISI247_1 |
| | tet39 | $bla_{MIR}$ | | aac(6′)-Ib_1 | | erm42 | | | | tnpA_4 |
| | tet44 | cphA_2 | | aadA5_2 | | mphA | | | | IS613 |
| | tetL_2 | $bla_{CMY\_2}$ | | aadA10 | | lsaC | | | | IncN_rep |
| | tetE | $bla_{SHV11}$ | | aac(6′)-II | | msrE | | | | Tp614 |
| | | $bla_{OXA10\_1}$ | | aacC2 | | | | | | ISAba3 |
| | | $bla_{CMY\_1}$ | | aadA7 | | | | | | IncQ_oriT |
| | | $bla_{MOX}/bla_{CMY}$ | | aphA3_1 | | | | | | Orf37- |
| | | $bla_{BEL}$-nonmobile | | | | | | | | IS26 |
| | | ampC_4 | | | | | | | | Tn5 |
| | | $bla_{CMY2}$ | | | | | | | | |
| | | ampC_1 | | | | | | | | |
| | | ampC_6 | | | | | | | | |
| | | $bla_{SHV\_2}$ | | | | | | | | |
| | | $bla_{ACT\_2}$ | | | | | | | | |
| | | $bla_{PER\_1}$ | | | | | | | | |
| | | $bla_{ACT}$ | | | | | | | | |

MLSB—macrolide, lincosamide, and streptogramin B; MGEs—mobile genetic elements.

### 3.1.3. Core Resistome

The core resistome analysis on the HT-qPCR dataset revealed 47 genes to be prevalent in at least 10% of the samples (Figure 2). The gene with the highest prevalence was *tetPA*, followed by *sul1_1*, *blaTEM_1*, *sul1_2*, and so on.

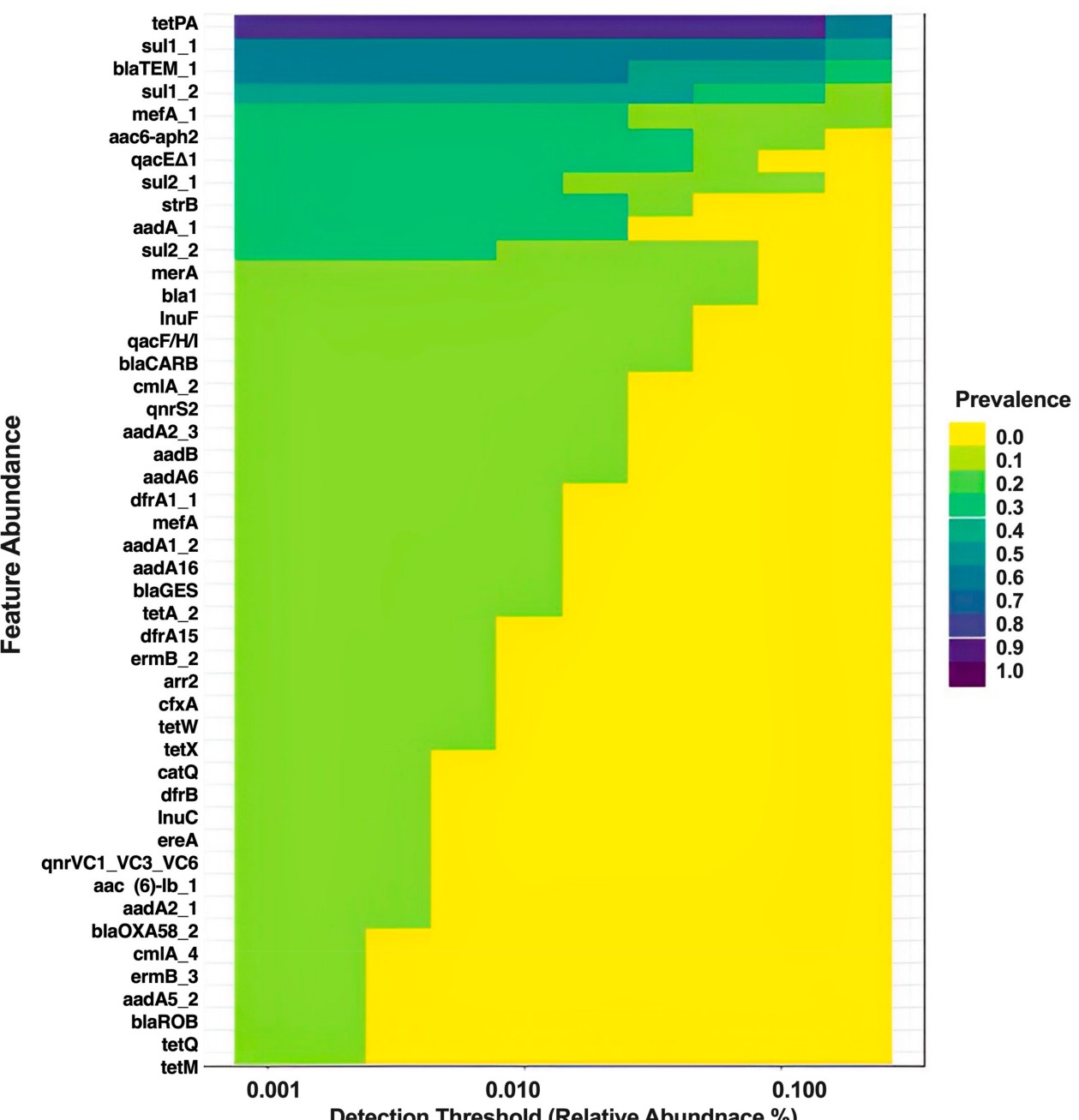

**Figure 2.** Core resistome identified using HT-qPCR method.

### 3.1.4. Taxonomic Profiles

The HT-qPCR panel possessed the markers for two bacterial families viz. Bacteroidetes and Firmicutes, as well as six pathogenic bacterial genera such as *Acinetobacter baumanii*, *Campylobacter*, *Enterococci*, *Klebsiella pneumonia*, *Pseudomonas aeruginosa*, and *Staphylococcus*. Amplification products were obtained only for the two bacterial families. The relative abundance (RA%) is shown in Table 2. The RA % for Bacterioidetes ranged between 0.13 and 1.27, whereas for Firmicutes it was between 0.11 and 0.55. None of the bacterial genera

were positive at any of the sites. The 16S rRNA gene showed an RA% of 1.00 in all the samples, suggesting the presence of other bacterial families/genera at these locations.

**Table 2.** Taxonomic profiles of bacterial domains observed through HT-qPCR.

| Gene | Site 1 | Site 2 | Site 3 | Site 4 | Site 5 | Site 6 | Site 7 | Site 8 | Site 9 | Site 10 | Site 11 | Site 12 |
|---|---|---|---|---|---|---|---|---|---|---|---|---|
| Bacteroidetes | 0.23 | 0.44 | 0.19 | 0.37 | - | 1.27 | 0.13 | 0.84 | - | - | - | 1.99 |
| Firmicutes | 0.14 | 0.08 | 0.27 | 0.11 | 0.14 | 0.55 | 0.23 | 0.13 | - | - | - | 0.29 |
| *Acinetobacter baumannii* | - | - | - | - | - | - | - | - | - | - | - | - |
| *Campylobacter* | - | - | - | - | - | - | - | - | - | - | - | - |
| *Enterococci* | - | - | - | - | - | - | - | - | - | - | - | - |
| *Klebsiella. pneumoniae* | - | - | - | - | - | - | - | - | - | - | - | - |
| *Psuedomonas aeruginosa* | - | - | - | - | - | - | - | - | - | - | - | - |
| *Staphylococci* | - | - | - | - | - | - | - | - | - | - | - | - |
| 16S rRNA | 1.00 | 1.00 | 1.00 | 1.00 | 1.00 | 1.00 | 1.00 | 1.00 | 1.00 | 1.00 | 1.00 | 1.00 |

*3.2. Comparative Analysis of HT-qPCR with Shotgun Metagenomic Sequencing*

Both the HT-qPCR and SMS methods identified ARGs, MGEs, and integrons in the collected specimens. The number, types, and class of genes discovered with SMS were much higher than HT-qPCR [28]. Detailed descriptions of the number of genes and drug classes identified using the two methods are provided in the following text. The pros and cons of both methods have been discussed as well.

3.2.1. Gene Numbers/Richness

While the HT-qPCR method picked 100 ARGs, the SMS method captured gene sequences of 402 ARGs to be distributed among 12 sediment samples (Table S2). A total of 5 integrons (classes I, II, and III) were captured using the HT-qPCR method, whereas the SMS filtered 168 class 1 integron (*intI1*) sequences from the integral database. About 18 MGEs were identified with the HT-qPCR assay. These included transposons, plasmids, and insertion sequences. In the SMS approach, the sequences aligned against the plasmid finder returned 1567 matches. The log counts of each gene category are presented in Figure 3. The average number of ARGs using the HT-qPCR method per site was 11.58 as compared to 103.17 using SMS. The mean number of integrons was 0.92 and 21,341.58 using HT-qPCR and SMS methods, respectively. On average, 3.50 MGEs were recorded with HT-qPCR per site, whereas 23,597.08 were captured with SMS.

3.2.2. Core Resistomes

Contrary to the HT-qPCR resistome, the SMS profile was much more comprehensive, with >300 genes present in 10% of samples. The most dominant gene was *patA*, followed by *mexN*, *adeF*, *AcrF*, *TaeA*, etc. (Figure 4).

3.2.3. Drug Classes

The gene classes recorded with the SMS method were more in numbers (28) and diversity. In agreement with the HT-qPCR method, beta-lactams (cephalosporins. Cephamycin, carbapenem, penam, penem, and monobactam) dominated the gene classes (37%). This was followed by macrolides (19%) and tetracycline (7%). The remaining gene classes were fluoroquinolone (6%), lincosamide (4%), phenicol (4%), streptogramin (4%), peptide (3%), glycopeptide (3%), aminocoumarin (3%), diaminopyrimidine (trimethoprim) (2%), rifamycin (2%), glycylcycline (1%), acridine dye (1%), triclosan (1%), pleuromutilin (1%), sulfonamides (1%), and fosfomycin (1%) (Figure 5a). In addition to these, low-abundance genes (<1.0%) such as Nucleoside, Oxazolidinone, Fusidic acid, Mupirocin, Elfamycin, Nitroimidazole, Antibacterial free fatty acids, and Nitrofuran were also picked with the SMS method (Figure 5b).

The HT-qPCR is limited in capturing the low copy numbers of genes (<10). This is most likely the reason for null amplification at sites 6, 9, 10, 11, and 12 (Figure 5c). With the SMS method, genes belonging to these drug classes were observed at all the sites (Figure 5d),

suggesting the ability of SMS to capture even the low-abundance gene, where less than ten copies of genes were present. As evident from Figure 5d, many genes exhibited resistance against multiple drugs. This was missed with the HT-qPCR assay.

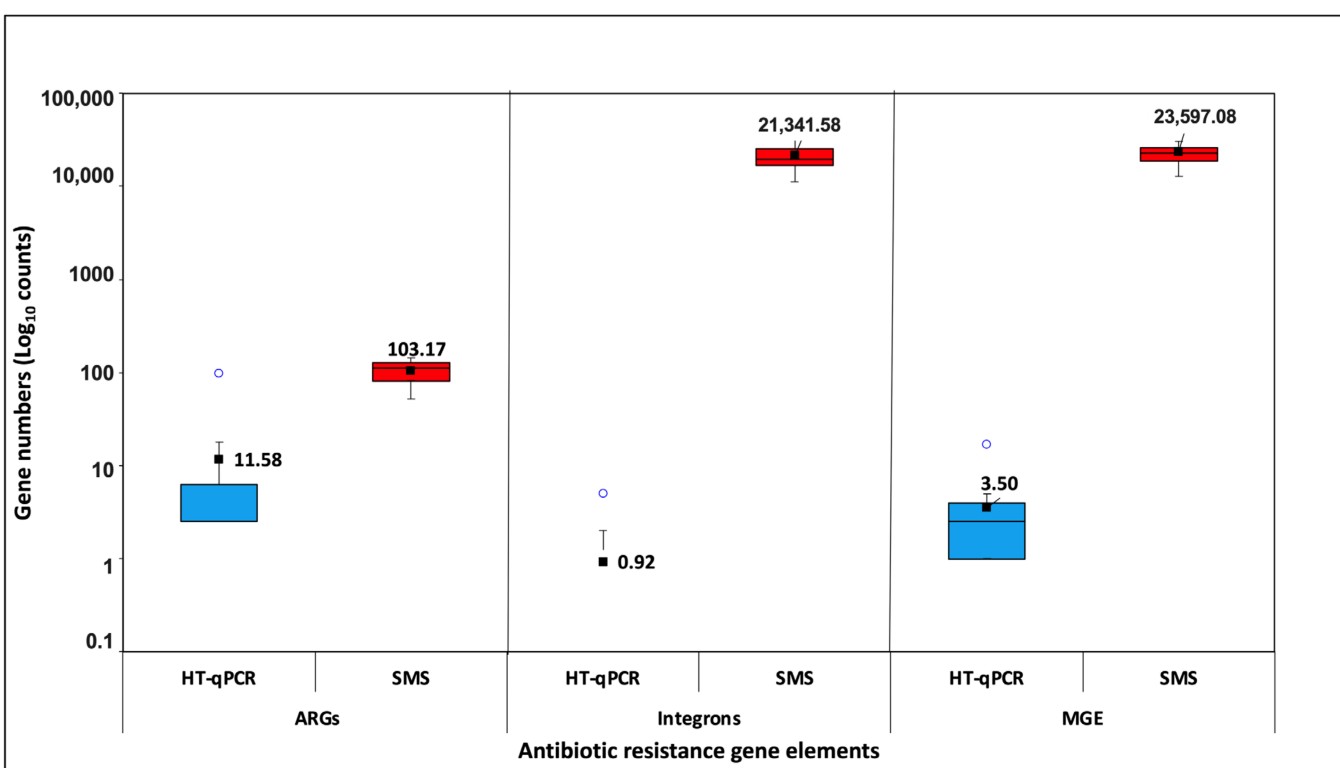

**Figure 3.** Box plot showing the distribution of resistomes in marine surface sediments. The blue boxes represent the HT-qPCR method, and the red boxes represent the SMS approach. The black square shows the average values of each gene type. The whiskers represent the standard deviation, whereas the blue balls are the outliers.

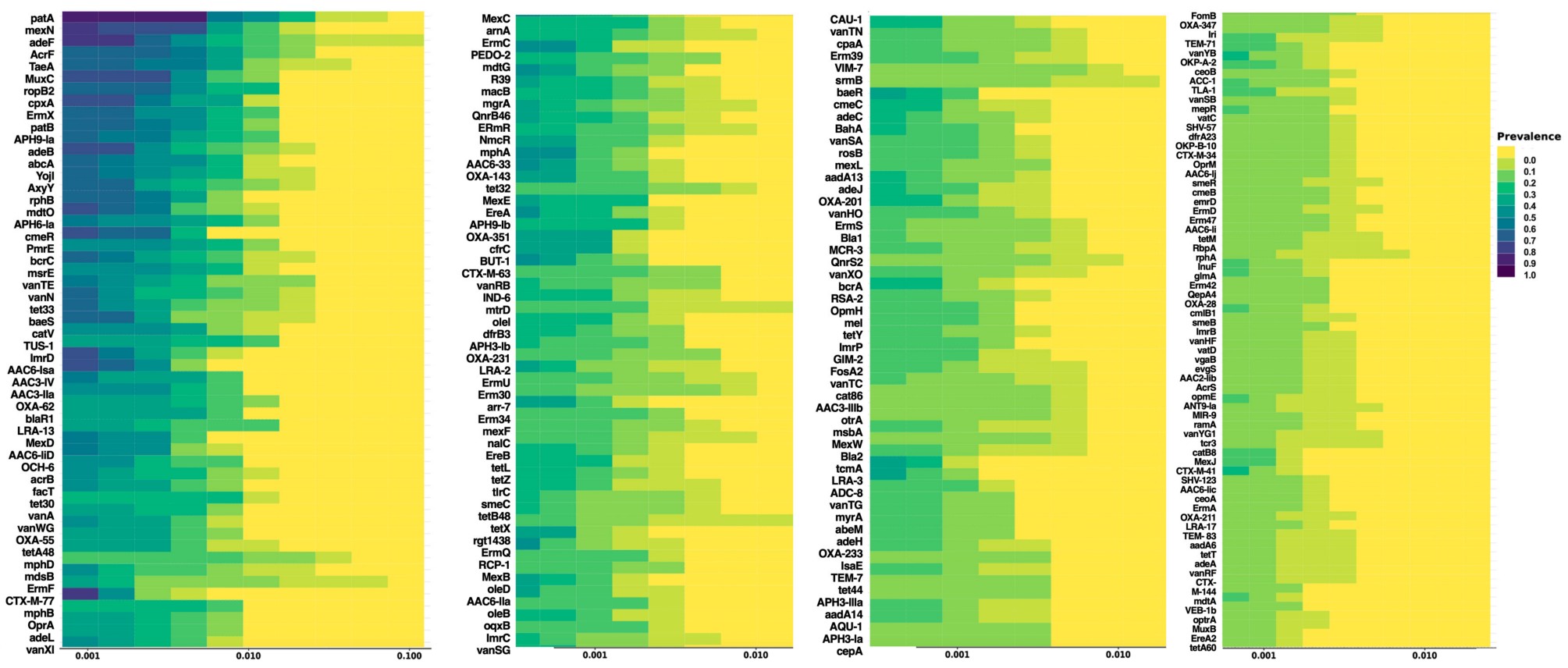

**Figure 4.** Core resistome identified with the SMS method.

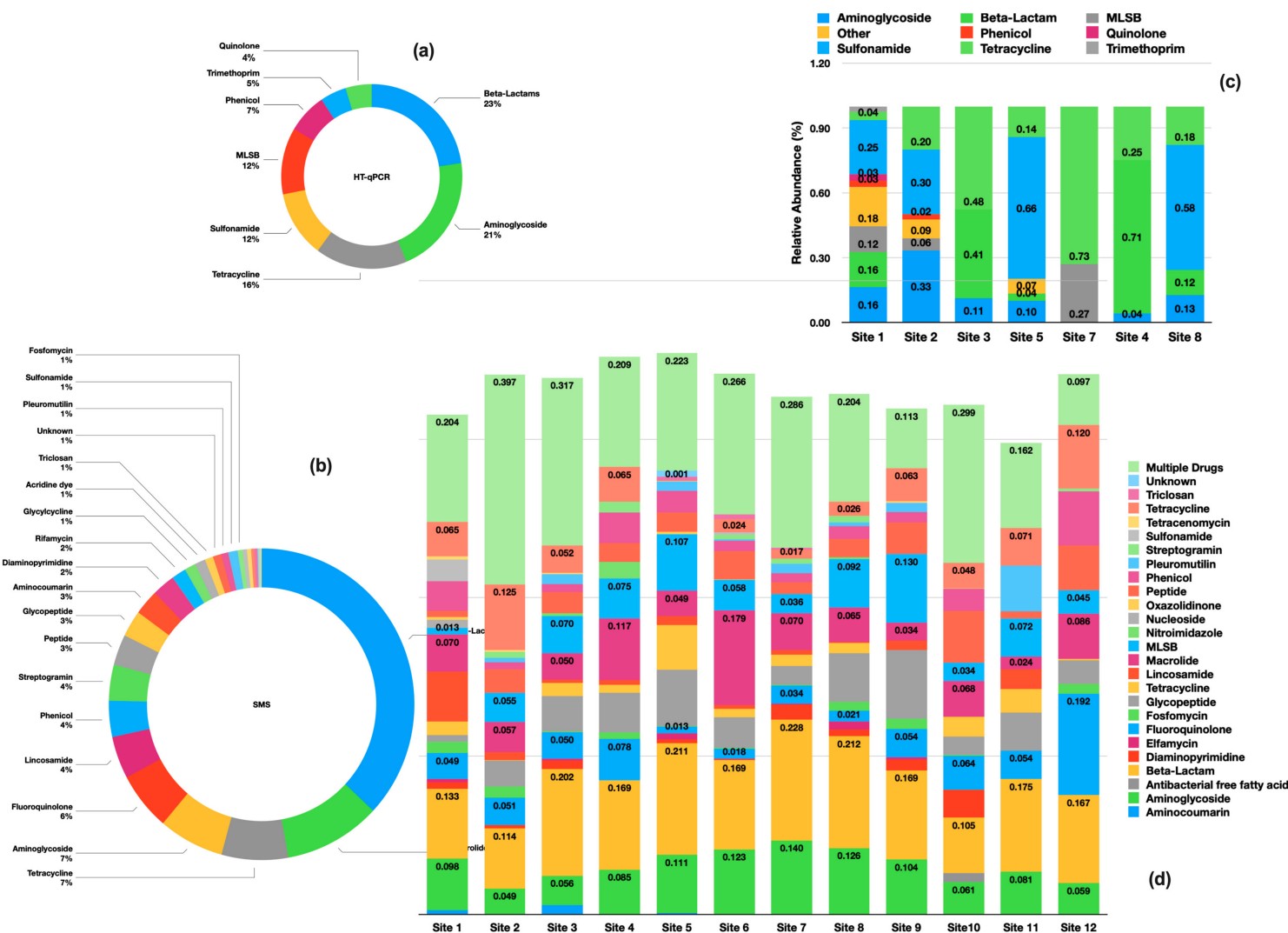

**Figure 5.** Drug classes identified with high-throughput molecular methods (**a**) HT-qPCR and (**b**) SMS. (**c**) Relative abundances of drug classes identified with HT-qPCR. (**d**) Relative abundances of drug classes identified with SMS.

### 3.2.4. Taxonomic Profiling via Shotgun Metagenomic Sequencing

Unlike the HT-qPCR assay, the taxonomic binning of the predicted ORFs in the SMS method captured the presence of all the microbial domains (~87%). Some other communities (12%), most likely the higher organisms, were also found in the present samples. Among the microbial population, overwhelming abundances of bacteria (86%) were recorded whereas only 1% were Archaea (Figure 6a) and <0.001% were viruses as well as Eukaryota. The number of sequences affiliated with each bacterial phyla in the sediment samples exhibited a dominance of Proteobacteria (Min 42% at S4 and Max 59% at S8) followed by Bacteroidetes (Min 6% at S12 and Max 18% at S3), Cyanobacteria (Min 1% at S5, S12, S1, S6, S10, and S8 and Max 10% at S2), Actinobacteria (Min 0.3 at S1 and Max 6.5% at S10), and Firmicutes (0.0–4%) (Figure 6b). The selective primers set in the HT-qPCR only amplified Bacteroidetes and Firmicutes DNA. The dominant genera were *Woeseia* (Min 0.2% at S1 and Max 10.6% at S11), *Marinobacter* (Min 0.1% at S5, S8, S7, S3, S11, and S10 and Max 12% at S1 and S6), *Pseudomonas* (Min 0.1% at S5, S7, and S11 and Max. −6.5% at S6), *Vibrio* (Max 4.7% at S2), *Loktanella* (Max. 3.8% at S6), and *Rubrivirga* (Max. 3.6% at S4), (Figure 6c). Among the prevalent genera, species of *Vibrio* and *Pseudomonas* are pathogenic. In addition to these ESKAPE (E-*Enterobacter* sps.; S-*Staphylococcus aureus*; K-*Klebsiella pneumonia*; A-*Acinetobacter baumanii*; P-*Pseudomonas aeruginosa*; and E-*Enterococcus faecium*) pathogens were found in very low abundances (Figure 6d). None of these pathogenic forms or ESKAPE genera were amplified with the HT-qPCR primers. ESKAPE are the WHO-listed priority genera that call for immediate attention while monitoring AMR.

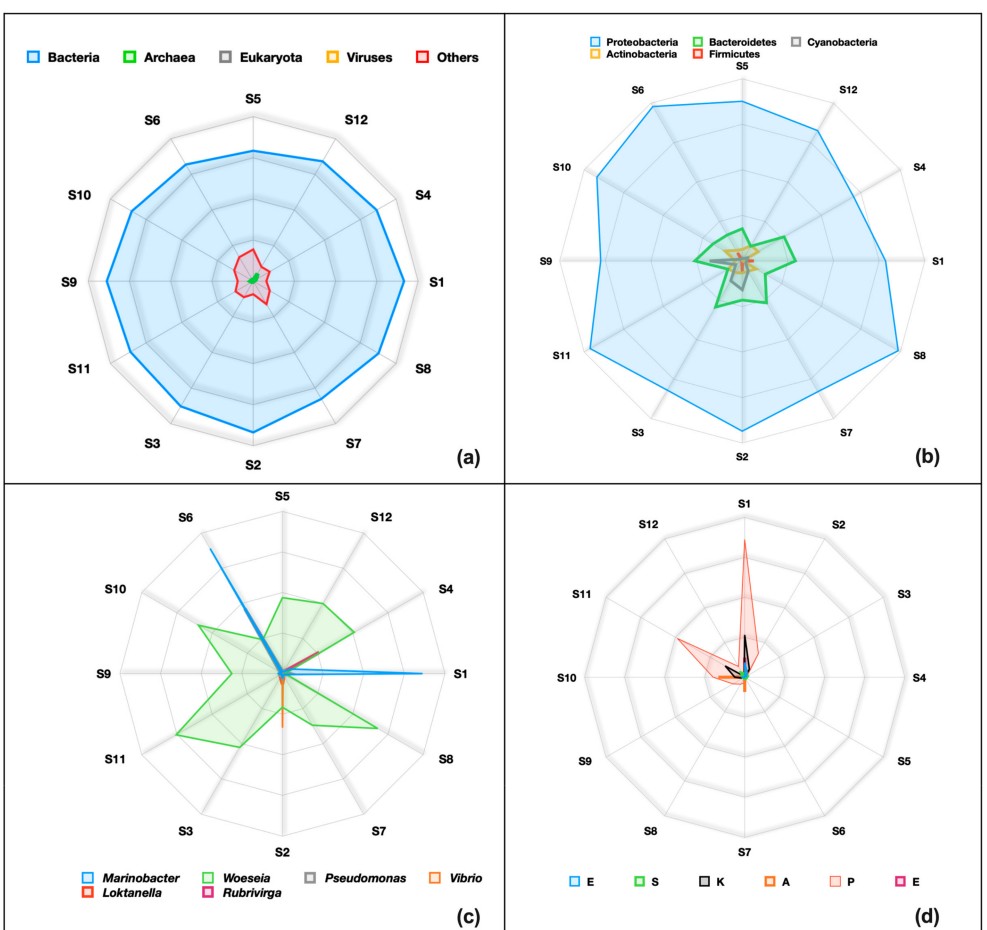

**Figure 6.** (**a**) Major domains, (**b**) phylum, (**c**) genera, and (**d**) ESKAPE pathogens found in marine sediments collected from Kuwait.

## 4. Discussion

Molecular methods successfully captured the antibiotic resistance gene elements (ARGEs) in the present study. Owing to the diverse nature of marine ecosystems, the SMS method was more comprehensive and informative compared to HT-qPCR. Our results completely agreed with the resistome profiles of a freshwater reservoir in Xiamen, China [28], mapped through similar approaches.

### 4.1. Target Genes and Primers in HT-qPCR

The primer sets in HT-qPCR targeted ARGs from 11 major classes including aminoglycosides, amphenicol, beta-lactams, sulphonamides, trimethoprim, multidrug resistance (MDR), florfenicol, MLSB, tetracycline, quinolone, and vancomycin. Of these genes, eight families were detected in at least one location. The main reason behind the fewer ARGEs captured with HT-qPCR is the limited number of primers available for target genes. The primers are designed based on the genes accessible in the common databases of CARD [49], ResFinder [50], Argannot [51], Megares [52], Plasmidfinder [53], Integral [54], and Isfinder [55]. Although novel genes are being added to these databases at a rapid pace strengthening the feasibility of additional primers, however, the time taken for validation of primers should be appropriated. The SMS method, on the other hand, does not depend on any primers, rather the complete sequences of the environmental DNA at a reasonably high depth are practically viable for richer ARGE descriptions. The same justification applies for the taxonomic profiling.

### 4.2. PCR Biases and Artefacts

PCR biases are common in qPCR assays [56]. It is important to give a thought to the denaturation and annealing regimes for multiple primers in the same assay. In the present study, we simultaneously amplified 296 genes, and it is quite possible that the conditions optimum for a group of primers might not be amiable for others. The SMS method on the other hand did not involve any PCR step except for the adapter and index ligation procedures. These steps involve very few cycles and hence the chances of amplification biases are rare. Similarly, higher $C_T$ values may not be representative of environmental concentration as amplification bias exists [57].

### 4.3. Low Abundance

Environmental samples are challenging regarding the biological yield [57–60]. Among the aquatic environments, marine sediments are vulnerable as co-precipitation of inhibitory substances makes DNA recovery difficult [57,58]. In the present samples, very low DNA yields were obtained (30–998 ngs) [46]. Although multiple isolation was conducted and a pool of DNA samples was used, genes with a low abundance of less than 10 copies were still missed in the HT-qPCR assay [61]. The sequencing by synthesis chemistry in the SMS application and paired-end sequencing at $2 \times 150$ bp generated 6 GB of data per sample. This sequencing depth was satisfactory for capturing genes with RA% as low as 0.001%. However, increasing the sequencing coverage by up to 12 GB of data per sample would be more reliable in predicting the rare genes [62]. The low abundance of pathogenic microbes in marine environments corroborates the few microbial genes captured using the HT-qPCR assays as compared to the significantly diverse microbial community composition with the SMS method [44].

### 4.4. Costing and Skills

Although the SMS method is considered cost-intensive, in the present study, surprisingly, the per-sample cost of HT-qPCR was USD 875, whereas the sequencing price was USD 320. An in-house qPCR assay for mapping ARGs from aerosols was also conducted at a comparable price [63]. In yet another study, the SMS was conducted for USD 200 for 10 Gb of data [28]. It is also important to note that the sequencing prices include the bioinformatics analysis. Prices vary depending on the service providers, the platform

used, and even the country where these facilities are available. Shipment costs of kits and consumables are also region- and country-bound. The SMS prices are based on sequence providers located in India, China, and Korea, whereas the HT-qPCR costs were obtained from some European countries. A higher skill level was still required for data interpretation and secondary analysis of SMS results. Online freely available visualization software such as MicrobiomeAnalyst [47,64,65] and ResistoXplorer [47] are quite user-friendly and easy to use. The total time taken to analyze HT-qPCR was superior as the results were obtained on the same day; however, the sequencing data were generated in about four weeks' time.

## 5. Conclusions

We noticed differences in the types and concentrations reported with the HT-qPCR and SMS methods. Considering the reduced cost and ease of availability of analytical software, the SMS approach is a more comprehensive tool for environmental monitoring and surveys if the data processing time is not a constraint. However, if the data are required quickly, HT-qPCR certainly has an edge over SMS.

**Supplementary Materials:** The following supporting information can be downloaded at https://www.mdpi.com/article/10.3390/app132011229/s1, Table S1: Abundances of gene ids revealed by HT-qPCR assay; Table S2: Abundances of gene ids revealed by SMS approach.

**Author Contributions:** Conceptualization, N.H. and S.U.; Formal analysis, M.B., H.A.A.-S., M.K. and F.Z.; Funding acquisition, N.H. and S.U.; Investigation, S.U., M.K. and N.A.; Methodology, W.A.-Z., N.A., A.S. and F.Z.; Project administration, M.B. and H.A.A.-S.; Resources, S.U., W.A.-Z. and M.B.; Software, N.H. and A.S.; Validation, H.A.A.-S. and A.S.; Visualization, N.H.; Writing—original draft, N.H.; Writing—review and editing, N.H. and S.U. All authors have read and agreed to the published version of the manuscript.

**Funding:** This study was supported by the Kuwait Institute for Scientific Research grant no. EM123C.

**Institutional Review Board Statement:** Not applicable.

**Informed Consent Statement:** Not applicable.

**Data Availability Statement:** All the data of the HT-qPCR are presented within the manuscript or the Supplementary Files. The sequences have been deposited to the National Centre for Biotechnology Information (NCBI) and can be accessed under accession number PRJNA819259.

**Acknowledgments:** The authors are thankful to the Kuwait Institute for Scientific Research for supporting this study.

**Conflicts of Interest:** The authors declare no conflict of interest.

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
