# Peer review of "A Comparative Assessment of High-Throughput Quantitative Polymerase Chain Reaction versus Shotgun Metagenomic Sequencing in Sediment Resistome Profiling"

_applsci, doi:10.3390/app132011229_

Round 1

Reviewer 1 Report

A Comparative Assessment of High Throughput Quantitative Polymerase Chain Reaction Versus Shot-Gun Metagenomic Sequencing in Sediment Resistome Profiling is a crucial and highly relevant article for researchers interested in profiling resistance and which tool to employ.

I recommend the following changes to the manuscript.

1.       What threshold are the authors referring to by the phrase “the thresholds are alarming”, please be specific.

2.       Definition of CDC should be revised.

3.       Figure 1 should be revised to improve the quality; font size should be increased to at least 12.

4.       Figure 2 should be revised to improve the quality, especially the pixel quality.

5.       The beauty of Figure 3 can be improved to enhance the manuscript.

6.       Figure 4 must either be removed entirely or be improved significantly. Readers cannot see what is presented. I recommend that the figure be split into 4 parts and then magnified so as to improve the figure quality.

7.       Figure 5c and 5d should be improved.

8.       Figure 6 should be improved. 

Reviewer 2 Report

Antibiotic resistance is the greatest threat to human health and environmental surveillance still remains the most neglected sector in the fight against AMR. The authors have compared two tools that are commonly used to monitor for ARGs in environmental surveillance and I support that the manuscript should be published. Nevertheless, I recommend few additions to add more value to the manuscript. The authors conducted metagenomic sequencing and HT-qPCR and found varying concentration of ARGs. Based on the concentration, it would add significant value to the manuscript if the authors could recommend which genes are appropriate to use for monitoring in Kuwait when using qPCR and funds are limited to support monitoring of `limited number of ARGs. Therefore please recommend in discussion which genes should be monitored in Kuwait when funds are enough to monitor lets say 5-6 gens using qpcr. which genes do you recommend? Also the abstract ranks the ARGs, maybe intrepret such findings of high beta-lactam resistant genes.

Specific comments are below:

Section 2.3. For HT-qPCR the ARGs were selected based on a study conducted in China. Any reason to follow the surveillance of same genes as in Sweden? HT-qPCR usually can be affected by low efficiency. Wasn't positive control added to form standard curves? What about NTC? Any specific reason to put cut off Ct value of 27? Did HT-qPCR involve pre-amplification step? Was a negative template control included to check for contamination during preamplification?

Line22. These ARGs were resistant to..... Please rephrase it. ARGs are not resistant but encodes for resistance to certain antibiotics. Also somewhere its ARGEs and somewhere ARGs. Please maintain uniformity as we have to minimize the use of such acronyms.

Line 345. How was 875 and 320 USD calculated. Doesn't need to be addded but am curious. Please can you share the costing analysis.

Line 357. Please add a comma after however.

English proficiency is fine. 

Reviewer 3 Report

  1. The paper titled “A Comparative Assessment of High Throughput Quantitative Polymerase Chain Reaction Versus Shot-Gun Metagenomic Sequencing in Sediment Resistome Profiling described the comparison of the high throughput quantitative polymerase chain reaction (HT15 qPCR) and shot-gun metagenomic sequencing (SGM) for the detection of antibiotics and resistance pattern. The manuscript has potential but need following changes before consideration
  2. Title is fine and describe the article in elaborative way
  3. Abstract is written good however describe methodology and results so revise abstract considering methodology, results and applications stakeholders for which this paper is useful at the end of abstract  
  4. Introduction is written good but there is a dire need for the addition of classification information for different antibiotics along with their impact on the substrate like animals and humans. Although there is still information for the plants and terrestrial invertebrates
  5. In material and methods, what was the criteria for selection of “Research sampling area like sediments” that need to be described in the manuscript in detail in a separate heading
  6. In material and methods, what criteria for the antibiotics of the samples analysis and sampling need to be defined in a separate headings
  7. In material and methods, “statistical analysis” should be provided in separate heading
  8. In Results do not write “ARGs and Drug Classes” & “Taxonomic profiling by SMS “as abbreviation and rather use full name in the heading
  9. In discussion, remove headings and focus on paragraph or follow the journal guidelines
  10. Conclusion is vague so modify as per research objective and support it with data findings
  11.  
  1. In the manuscript, grammatical mistakes observed on several places so there is need to go through the paper for language and grammatical mistake removal for better flow of language  
